# Uncovering Youth's Invisible Labor: Children's Roles, Care Work, and Familial Obligations in Latino/a Immigrant Families

Vanessa Delgado

Department of Sociology, Washington State University, Pullman, WA 99163, USA; vanessa.delgado@wsu.edu

**Abstract:** This paper examines Latino/a children's roles and obligations to their immigrant families. Bridging insights from the literature on the "new sociology of childhood," immigrant incorporation, and care work, this essay argues that children perform important—but often invisible—labor in immigrant families. Dominant ideologies depict childhood as an "innocent" time wherein young people are in need of guidance and are too underdeveloped to make meaningful contributions. However, this construction of childhood ignores the lived realities of the children of immigrants, who often serve as gatekeepers and connect their families to services and resources in their communities. This essay examines six dimensions of support that the children of immigrants provide to their families, namely, language and cultural help, financial contributions, bureaucratic assistance, emotional labor, legal support, and guidance with technology. This essay concludes with implications for scholars, students, and policymakers on the importance of recognizing this labor, along with future directions for research.

**Keywords:** children of immigrants; Latino/a families; immigrant families

## 1. Introduction

Sociologists have long established that contemporary ideologies of childhood render youth as being precious, emotionally priceless, and in need of protection (Zelizer 1994). Childhood as a social construct is perceived to be an "innocent" time, wherein young people are in need of guidance and are too underdeveloped to make meaningful contributions (James 2017; Orellana 2009). However, this celebration of youth's innocence and purity undermines the lived realities of children who must undertake roles and responsibilities to support their families. A growing number of studies demonstrate that children with marginalized social locations frequently take on tasks to buffer the structural inequalities experienced by the family (see Canizales and Hondagneu-Sotelo 2022; Delgado 2020a).

Racialized working-class children of immigrants experience various inequalities that position them to transcend the contours of normative childhood. For instance, a key feature of the early life of children of immigrants involves performing labor to facilitate the integration of their immigrant families. Such labor can include translating for immigrant parents in a variety of settings, including at parent–teacher conferences, doctor's visits, government service offices, colleges, and broadly helping immigrant parents to understand cultural references in everyday interactions; they may even provide financial support through monetary gifts or by working in family-run businesses (Orellana 2009; Katz 2014; Estrada 2019; Delgado 2020a; Flores 2021; Kwon 2014). As such, the children of immigrants serve as important social actors in their families and in broader society.

The purpose of this study is to examine children's roles and obligations in Latino/a immigrant families. In doing so, this essay has two goals. The first goal is to bring together three bodies of sociological literature that have typically explored family labor independently. Existing research on the "new sociology of childhood" acknowledges young people as social actors and highlights how the social location of youth shapes one's upbringing

(Matthews 2007). This research has been crucial to understanding how marginalized children contend with race, class, and gender in their social worlds. However, research on younger people's experiences within immigrant families remains scant. Additionally, the immigrant incorporation literature primarily examines Latino/a family integration from one generation to the next, with incidental discussions of young people's labor (Delgado 2020a). Lastly, the care work literature has traditionally focused more on women's and less on children's labor. In all, creating a corpus from these three bodies of literature helps provide a more complete theoretical underpinning of youth's labor in immigrant families. The second goal of this essay is to underscore the various dimensions of labor that children in immigrant families contribute to the household. Identifying this labor is important as much of this work often goes unnoticed in family research and society as a whole. This essay builds on the aims of the Special Issue by drawing attention to how young people's labor contributes to the integration and mobility of Latino/a immigrant families in the U.S. and abroad.

This review begins with an overview of the research within migration, childhood, and care-work studies as it relates to young people. It then moves on to discuss six dimensions of support that the children of immigrants provide to their families, namely, language and cultural help, financial contributions, bureaucratic assistance, emotional labor, legal support, and guidance with technology. This essay focuses on Latino/a families because Latino/as in the U.S. are disproportionally affected by immigration policies and racialized systemic violence (Menjívar and Abrego 2012; Canizales and Vallejo 2021). It concludes with the implications for scholars, students, and policymakers regarding the importance of recognizing this labor and future directions for research.

## 2. The New Sociology of Childhood and Migrant Youth

In the mid-1980s, the "new sociology of childhood" emerged as a lens to capture how children serve as social actors who are capable of shaping their families, communities, and broader society (Corsaro 2005; James et al. 1998; Matthews 2007; Prout 2011). This lens challenges the widespread notions of youth, including the perception that children are simply a "lump of clay in need of being molded" and are waiting to be socialized by adults (Knapp 1999, p. 55). A new wave of research within the sociology of childhood now recognizes the powerful role that children can play in society and examines how young people actively secure opportunities at school, engage in decision-making alongside parents, make sense of race both in and outside school settings, help their parents in the workplace, and foster relationships with others (Estrada 2019; Katz 2014; Lewis 2003; Orellana 2009; Prout 2005). Studies of childhood also pay considerable attention to research methodology, including how to collect and interpret data regarding children (Christensen 2004; Christensen and James 2000; Christensen and Prout 2002; Davis 1998; Fine and Sandstrom 1988; Greene and Hogan 2005; Hood et al. 1999; Matthews 2007; Orellana 2019). Indeed, allowing children to speak for themselves can create challenges as adult researchers may not interpret children's words accurately, study contents may create emotional distress for youth, and young participants may only share what adults "want to hear." Nevertheless, the "new sociology of childhood" encourages sociologists to take children seriously and reflect upon traditional methods to ensure that childhood studies are compassionate yet rigorous (Orellana 2019).

Children's labor for the family can differ based on social factors such as gender, class, and race/ethnicity. For instance, in "Growing Up Poor", historian Anna Davin (1996) argued that working-class children had quite different experiences than bourgeois children. Davin (1996) observed that working-class children have undertaken family and work responsibilities, even when youngsters are legally required to attend school. Conversely, bourgeois families often conceived childhood as a protected time wherein children ought to be free from duties. Gender also shapes these experiences, as girls—regardless of class background—are encouraged to undertake domestic care, including cooking, cleaning, and caring for younger kin (Archer 1992). Subsequent work has also made it clear that

race/ethnicity status only further complicates childhood experiences since children of color are exposed to racism and discrimination (Epstein et al. 2017).

In addition to class, gender, and race/ethnicity, scholars have observed that young people who grow up with immigrant parents undertake unique family responsibilities. For instance, scholars draw on terms such as "brokers," intermediaries", or "paraphrasers" to capture the process by which the youth bridge cultural, legal, and linguistic gaps between their immigrant parents and third parties (Delgado 2020b; Jones and Trickett 2005; Orellana 2009; Orellana et al. 2003; Tse 1995, 1996; Valdés 2014; Valenzuela 1999). This brokering role serves as an intra-family strategy, as families work together to overcome the barriers of living and adapting to a new country. Youth in immigrant families play a powerful role as they draw on their language skills, knowledge of popular culture, U.S. education, access to mentors, and experiential knowledge to help secure resources and opportunities, as well as to navigate challenges.

Young people's labor *within* immigrant families can also differ. Consider that immigrant families can occupy various legal statuses, ranging from protected, semi-protected, and unprotected (Delgado 2022a). Young people with citizenship can help their undocumented parents by driving their parents to appointments, connecting parents with extended kin via cellphones and technology, sharing memorabilia across borders, and looking into the process of sponsorship and the adjustment of status (Delgado 2022b; García Valdivia 2022; Getrich 2019) Relatedly, Delgado (2020b) argues that children of undocumented immigrants serve as "legal brokers," wherein the younger family members provide information about laws and policies, legal rights, and guidance on claims-making for their undocumented parents. Younger family members actively work alongside their undocumented parents to navigate barriers to social services, spatial immobility, economic hardships, increased surveillance, and threats of deportation (Abrego 2016; Bean et al. 2015; Capps et al. 2005; Dreby 2015; Vargas and Pirog 2016; Yoshikawa 2011).

## 3. Care Work in Immigrant Families

In the context of families, care-work scholarship examines the labor that family members provide to ensure that the family unit runs smoothly. Care work is defined as paid and/or unpaid work that develops the capabilities of recipients (England et al. 2002). Care work can take place in the family, in nonprofit institutions, and in both the private and public sectors of employment. Examples of care work range from childrearing, volunteering one's time to a cause, and working in occupations such as social work or caregiving. Family members engage in specific types of care work, including parenting, sibling care, and caring for older and aging parents (Delgado 2020c; Gerstel 2000; Grigoryeva 2017; Hochschild and Machung 2012; Schmalzbauer and Rodriguez 2022). Much of this research on care work centers on women, exposing how gender inequality seeps into the lives of families and places uneven burdens on mothers and other female family members. Surprisingly, much less research focuses on the unique care work demands that emerge when immigrants transition to a new country.

The existing research on care work among immigrant families primarily focuses on migrant women who travel to a different country to meet the economic needs of their family[1]. Transnational care work, or "caregiving across borders," is a gendered experience wherein mothers take on employment and are expected to perform the traditional caregiving responsibilities ascribed to women (Hondagneu-Sotelo and Avila 1997; Bruhn and Oliveira 2022; Schmalzbauer 2004). The focus on migrant mothers helps elucidate the juxtaposition of gendered norms about parenting and restrictive immigration laws; however, this focus on motherhood can also inadvertently overlook the important labor that other members of the family provide to their immigrant families. Shifting the focus to youth in immigrant families helps underscore the important contributions made by young people and resists narratives that depict children as passive members of society.

Herein, I connect insights into care work literature with the "new sociology of childhood" and immigrant incorporation research to underscore the important labor that chil-

dren perform in immigrant families. The research within the "new sociology of childhood" makes it clear that children play an active role in their families, communities, and broader society. Immigrant integration scholarship draws attention to how children in migrant families face additional challenges as they help to orient their parents to life in a new country. I argue that the children of immigrants perform important labor in their families (that is, care work) that helps their immigrant families run smoothly and secure important resources required for their integration. The care work that the children of immigrants undertake can differ, based on the family's social position. For instance, children may have to perform additional labor if their parent has a vulnerable immigration status compared to those children who grow up with documented parents.

## 4. Dimensions of Youth Care Work in Immigrant Families

To uncover the various forms of care work that children partake in to support their immigrant parents, I focus on six dimensions of support: language and cultural, financial, bureaucratic, emotional, legal, and technological. Figure 1 (see below) provides an outline of each type of support and offers examples to elucidate how this support may take place. I caution the reader to consider these dimensions of support as the major ways in which youth can help and not as an exhaustive and static list. Young people may employ one or more types of support when supporting family members. For instance, they may draw on their language skills to construct an email (i.e., language and technology and media support). Young people can also employ their language skills when requesting social services (i.e., language support and bureaucratic assistance), while those who have undocumented parents may consider how best to ask for services without endangering their family (i.e., language, bureaucratic, emotional, and legal support). The extent of the support that they offer depends on the needs of the family and the resources available to young people.

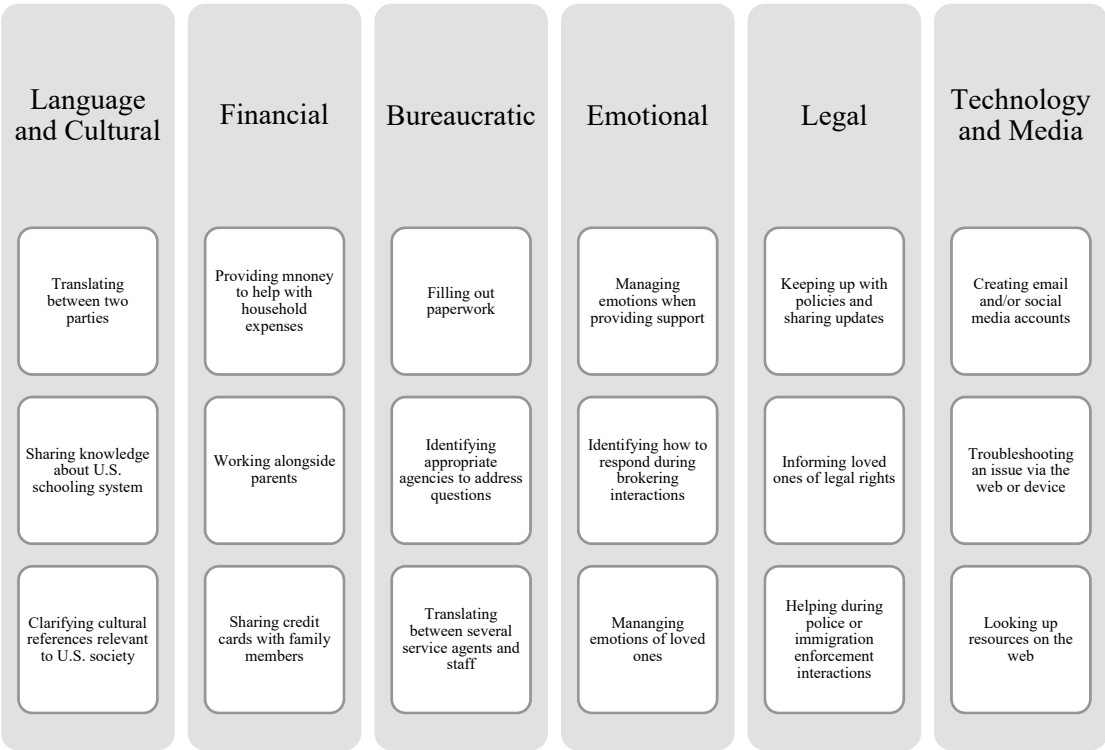

**Figure 1.** Dimensions and examples of young people's care work in immigrant families.

### 4.1. Language and Cultural Support

Scholars have established that the children of immigrants serve as language and cultural "brokers"—wherein they translate for or share resources with their immigrant kin (Kam and Lazarevic 2014; Tse 1996). Language or cultural brokering can include "[interpreting] and [translating] between culturally and linguistically different people and [mediating] interactions in a variety of situations including those found at home and school" (Tse 1996, p. 226). The published research on brokering has focused on how this brokering responsibility shapes educational outcomes (Borrero 2011; Buriel et al. 1998), the sense of belonging (Niehaus and Kumpiene 2014), attitudes about familial duties (Morales et al. 2012), and parent–child relationships (Buriel et al. 2006). While brokering can start between the ages of eight and ten (Buriel et al. 1998), there is research suggesting that brokering is a responsibility that transcends into adolescence and adulthood. Gender can also affect brokering patterns as daughters and young girls may do more labor in their families (Weisskirch 2010).

In adolescence, young people can become more aware of the hardships their parents face as immigrants—which can affect the brokering that they perform for their parents. For instance, Guan et al. (2016) find that youth who provide linguistic help report more compassion for their parents' struggles. Brokering can position young people to witness moments wherein their parents become the targets of racism, xenophobia, and nativism. Children can also learn about family mishaps, such as money problems. Indeed, Kwon (2014) finds that children who learn about their parent's financial problems through language brokering can experience a "hidden injury of class." Reynolds and Orellana (2009) note that children grapple with decisions to protect their parents during tense interactions. Young brokers do their best to help, but their status as "children" can sometimes limit how much help they can provide to their parents.

### 4.2. Financial Support

Research suggests that parents financially support their children at a greater rate than previous generations (Rumbaut and Komaie 2010); however, support is often reciprocal in Latino/a immigrant families. When compared to native-born families, Fuligni and Pedersen (2002) find that the children of Latino immigrants report a greater sense of obligation to the family and are significantly more likely to provide financial assistance to their parents and siblings. Lanuza (2020) notes that the children of Latino/a immigrants are more likely to provide monetary support to their parents during the transition to adulthood than white young adults. Similar trends have been found among upwardly mobile Mexican-Americans, wherein one-fifth of respondents reported that they "fully supported" their immigrant parents (Vallejo and Lee 2009). Family obligations may be contingent upon the parent's socioeconomic standing. Latinos with parents who face economic hardships report feeling more responsibility to help, compared with those whose parents are "doing just fine" (Vallejo 2012). Other work highlights that Latino/a young people work alongside their parents in both formal and informal employment (Estrada 2019). This research indicates that "giving back" and providing economic support to families can be a major aspect of working-class Latino/a youth's lives (Canizales and Hondagneu-Sotelo 2022).

Financial support also extends beyond the U.S., as scholars have documented that young people who migrate to the U.S. as unaccompanied minors send remittances to parents and other family members. Unaccompanied youths—many of whom are undocumented Latino/a teenagers in low-wage jobs—feel a sense of urgency to work and send money to loved ones (Canizales 2021, 2022). This urgency to work can position undocumented teenagers to take on exploitative jobs that often hire young people "on the spot." With a job, unaccompanied youths are able to send remittances and contribute to their family's survival abroad. For instance, Heidbrink (2018) notes that unaccompanied minors use their remittances to help pay off land debt, purchase food and clothing, and help younger siblings attend better schools. The transitional financial urgency and ex-

ploitive work environments negatively impact the health of undocumented youth workers (Canizales 2022).

### 4.3. Bureaucratic Support

Immigrants frequently encounter bureaucratic institutions such as government agencies or commercial businesses with highly regulated operating procedures. These bureaucratic institutions range from schools, hospitals, and clinics to social services offices. Interaction with such institutions can cause immigrants to seek out help from their bilingual children because following the administrative system or the rules of the organization may be a multi-step process. One such example is setting up a medical appointment or follow-up visit with a doctor. For instance, Katz (2014) finds that children serve as ad hoc translators for their parents during doctor's visits. Katz (2014) noted that even when translators are available, parents still rely on their children for support because professional translators can rush through information and even be dismissive of immigrant patients. Similar trends were reported during the COVID-19 pandemic, wherein the adult children of immigrants felt the need to step in during medical appointments to ensure that their parents received proper care (Delgado 2022c). Other research also notes that parents can turn to their eldest child to help them navigate school-related issues, such as enrolling in college or setting up an email address (Delgado 2020c; Flores 2021; Schmalzbauer and Rodriguez 2022). Young people and their parents work together to access resources and services when the bureaucratic process is convoluted, or when service agents are unsupportive. Youths with undocumented parents may face additional pressures when navigating bureaucracies because some undocumented family members may not want to engage with "surveilling" institutions that keep formal records (Asad 2020; Desai et al. 2020).

### 4.4. Emotional Support

Brokering interactions position the children of immigrants to engage in emotional labor—that is, the regulation of emotions and expressions (Hochschild 2015). Emotional labor includes the paid or unpaid—often invisible—work that one performs to make others comfortable (Hochschild 2015). When translating written material or oral conversations, youths must first absorb the information and then decide upon the best way to communicate its message to their parents. The level of emotional labor deployed can differ, based on the types of interactions that take place. For instance, young people may not fully engage in emotional labor if they translate in "low-stakes" scenarios, such as inquiring about the cost of an item at the store or when ordering a meal at a restaurant. However, certain brokering interactions require young people to engage in emotional management for themselves and their parents.

Existing research suggests that three factors prompt youths to employ emotion management techniques when brokering. First, if the young person is translating information that is sensitive or potentially damaging to their parent, they must decipher how to best translate such information in a delicate manner. For instance, in her interviews with the adult children of immigrants, Delgado (n.d.) finds that youngsters often engage in strategies to protect their parents' emotions. In one case, a young man recalls the time that he had to translate to his pregnant mother that her fetus, his potential brother or sister, no longer had a heartbeat and had miscarried. In this interaction, he struggles with his own emotions when processing the loss of his own sibling, while simultaneously worrying about how his mother would react to the devastating news. Second, if the young person feels dismissed—either by a third party or by their parent—negative emotions can arise about their translator role. Lastly, a negative sociopolitical context leaves room for young people to experience microaggressions, racism, and xenophobia during their brokering. Young people may feel angry or upset when racist remarks are made toward their parents (Crafter and Iqbal 2022; Delgado 2020a). At the same time, young people can also develop an empathetic stance toward their parents as they witness the hardships and struggles that immigrants face in the U.S. (Estrada 2016). Younger children may feel ill-equipped or



uncomfortable with responding to microaggressions, while older youths may step in and confront those third parties who disregard their loved ones (Delgado n.d.).

### 4.5. Legal Support

The children of undocumented immigrants are known to navigate the contours of legality and illegality in their families. For instance, undocumented youths must decide whether it is safe to disclose their family's undocumented status in a variety of settings, assess the risks in high-stakes situations (i.e., passing a driving under the influence [DUI] checkpoint), and dismantle harmful narratives regarding "deserving" and "undeserving" immigrants (Negrón-Gonzales 2014). Undocumented youths involved in political movements that advance the rights of immigrants regularly share legal resources and information with their undocumented parents. Undocumented college students who attend universities with targeted programming can also gain legal knowledge from "Know Your Rights" workshops and other on-campus events for undocumented students and will actively share this knowledge with their parents (Delgado 2022a, 2022b).

Youths may provide different types of support, based on their own legal protection. For instance, U.S.-born citizens can attempt to adjust the immigration status of their undocumented parents, may travel back and forth to their parents' home country, and take on tasks that are deemed "safer" for citizens (i.e., driving past immigration checkpoints) (Abrego 2019; Delgado 2022b; García Valdivia 2022; Getrich 2019). Youths with Deferred Action for Childhood Arrivals (DACA) status can also help their undocumented parents in many more ways than those who are not DACA-documented. DACA provides a two-year renewable work permit and temporary protection from deportation[2]. For an estimated 800,000 undocumented youths, "access to work" authorization allowed them to participate fully in the labor force, be accepted for better jobs, earn better wages, access driver's licenses, and matriculate into higher education (Patler et al. 2021b; Wong et al. 2017). In many families, DACA holders become the only ones with a state ID and a social security number—allowing them to drive and secure credit cards that their parents could benefit from. Abrego (2018) found that undocumented young adults with DACA were able to provide more financial support for their parents. Relatedly, Aranda et al. (2021) assert that DACA-documented youths emerge as "institutional brokers". DACA recipients are also able to use this role to stand up and engage in claims-making on behalf of or alongside their parents (Aranda et al. 2021). Existing research suggests that DACA may allow qualified undocumented youths to support their undocumented parents in similar ways to that of citizens. However, scholars caution that the prolonged uncertainty of DACA may undermine its benefits and negatively affect DACA-documented young adults (Gonzales et al. 2019; Patler et al. 2019, 2021a). As such, the mental toll of performing legal brokering can differ between citizens and DACA-documented youth (Delgado 2022b).

### 4.6. Technology Support

In the last two decades, technology and media have grown significantly. Bureaucratic institutions that provide important goods, such as medical care, education, and social resources, have adopted technology to facilitate or ease the accessibility of their services to community members. For instance, many K-12 schools rely on online applications to admit new or transfer students. Parents also can track their children's academic progress via an online portal that provides information about school requirements, the student's grades, and their attendance history (Olmstead 2013). While some studies demonstrate that technology can help increase parental involvement (Hollingworth et al. 2011; Olmstead 2013; Patrikakou 2016), immigrant parents can face unique barriers to accessing technology.

The majority of low-income immigrant families are under-connected to the internet and usually rely on mobile devices with data limits to surf the web (Rideout and Katz 2016). Immigrants may also struggle to navigate technology and material that is primarily written in the English language (Gu 2017). Consequently, Katz (2010) observed that the children of immigrants serve as "media brokers" in their families. Youth can provide

technological support that ranges from reading and translating online webpages and creating email accounts to assessing whom to receive services from in their community (Katz 2010). The COVID-19 pandemic increased the demand for technology services, given that these services were no longer available in person. The increased demand for technology positioned the children of immigrants to support their parents through Zoom meetings, email requests for services, telehealth, and creating appointments for personal care (Delgado 2022c).

## 5. Directions for Future Research

The existing research on youth labor makes it clear that children in immigrant families can provide a range of support. This support is essential to receiving services, learning new skills, and securing opportunities for mobility. Further research is required to better understand how social locations, such as ethnic differences, place, and aging, shape the labor that youth provide to their families. Below, I provide suggestions for future research in an effort to enrich our understanding of young people's roles in family work.

Very few studies examine young people's labor among indigenous immigrant families—creating fertile ground for future research. Most studies on Latino/a young people's translation or brokering support focus on English–Spanish interactions. However, it is important to consider that the umbrella term of "Latino" or "Hispanic" can minimize the rich ethnic and linguistic complexities among this ethnic group, including indigenous Mexican and Central and South American migrants. There are an estimated 15,000–19,000 individuals from Latin America who speak an indigenous language in the U.S., ranging from Zapotec, K'iche', and Mayan, among others (Comunidades Indigenas en Liderazgo (CIELO) (2022)). Language and ethnic barriers can make integration harder for indigenous migrants, which may position young people in these families to broker interactions in their native language, Spanish, and English. Resources for these families are likely to be limited because Spanish is presumed as the dominant language for all Latino/a migrants in the U.S.

Studies should also consider the significance of place in young people's care work. Much of the research on brokering has focused on major U.S. cities, many of which are traditional immigrant gateway locations. Future work should assess how living in towns or cities with a limited supportive infrastructure for immigrant Latino/a migrants shapes the labor that youths perform for their families. The few studies on translation or brokering support in predominately White public spaces suggests that youths may feel a heightened awareness of their racial/ethnic background or experience racial microaggressions when performing their labor (Reynolds and Orellana 2009). It is also possible that youths may be called upon to perform more labor because institutions lack the resources to provide equitable services for these populations. Future research should assess how location and place shape the labor needs of youths in immigrant families.

Finally, a focus on the transition to adulthood and aging can illuminate how self-development impacts youth support to the family. While the contributions of child brokers cannot be overstated, it is important to recognize that young people's brokering may evolve as they enter various developmental stages later in their life. Adulthood opens up the opportunity for youths to broker in different and more sophisticated ways. Upon turning 18, young people are recognized as independent adults. These young adults are now able to vote, perform jury duty, complete their taxes, sign contracts, enroll in the military, get married, and make decisions without the consent of their parents. Neurocognitive and psychology literature uncovers how brain maturation over time is linked to increased cognitive and reasoning skills (Luna et al. 2004). Compared to their childhood, young adults develop a broader vocabulary, sharper critical-thinking skills, improved self and social awareness, and a stronger presence during brokering interactions. However, little is known about how entry into emerging adulthood provides young people with new tools to employ when brokering for their immigrant parents. Such research is required to elucidate how Latino/a immigrant families work together to navigate key social institutions in the US—and how the brokers' positionality as adults shapes their access to resources.

## 6. Discussion and Conclusions

Immigrant families work together to navigate the barriers that arise when moving to a new country. As such, children draw on their language abilities and understanding of U.S. norms to facilitate access to key actors, resources, and important services. This labor is often considered a "normal" characteristic of growing up with foreign-born parents; to be clear, this labor should not be considered abnormal because it deviates from the traditional construction of Western childhood. However, by characterizing this labor as "normal," we undermine the social structures that position young people to engage in these helping strategies. Youths emerge as brokers because the U.S. does not have the structures in place to support the integration of their immigrant parents.

Policies are required to help immigrant families to integrate into U.S. society, particularly as the U.S. moves toward a post-COVID-19 society. The COVID-19 pandemic exacerbated the labor that children of immigrants perform for their parents. For instance, early studies suggest that older siblings stepped up to help their younger kin with remote learning by providing access to hot spots, translating between parents and teachers, helping parents to learn how to create Zoom and email accounts, and purchasing internet access and computers (Delgado 2020c). Youths also helped their parents navigate healthcare services that transitioned to the web. Immigrant families were forced to work together as they navigated telehealth and circumvented the no-visitor rule (then, later, the one-visitor rule) in hospitals and clinics (Delgado 2022c). Other children, particularly those with undocumented parents, may have had to step up more because their parents were ineligible for service programs, such as the federal Economic Impact Payments scheme (i.e., federal stimulus checks). Policies that support the integration of immigrants will relieve youths from shouldering the responsibility to amalgamate their parents with U.S. society.

Latino children are one of the fastest-growing groups in the U.S.; by 2050, they are projected to make up nearly one-third of the child population (Mather 2016). About half of these children have at least one parent who was not born in the U.S. (Clarke et al. 2017). Many of these youths are tasked with helping their immigrant parents to navigate key social institutions in the U.S. However, this brokering role can be challenging for young people as it adds additional pressures to their childhood. Uncovering the various dimensions of this "invisible" labor that the children of immigrants provide to their families highlights the need for state and federal policies to intervene and engage in efforts to reduce the inequalities faced by immigrant families.

**Funding:** This research received no external funding.

**Data Availability Statement:** Not applicable.

**Acknowledgments:** A special thank you to the editors of this Special Issue, Joanna Dreby and Leah Schmalzbauer, for their careful feedback on this paper. The paper also benefited tremendously from my colleagues' feedback during the "Rethinking the Mobilities of Migrant Children and Youth across the Americas" symposium. The anonymous reviewers provided helpful comments on the final version of this manuscript.

**Conflicts of Interest:** The author declares no conflict of interest.

## Notes

[1]  There is research on immigrants in care work professions. This essay draws attention to care work research within the family unit.

[2]  The Fifth Circuit Court of Appeals issued a decision on the legality of the DACA program on 5 October 2022. Current DACA recipients are able to continue their benefits and renew their work authorization. However, first-time applications will not be processed at this time (National Immigration Law Center 2022).

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
