# Peer review of "Uncovering Youth’s Invisible Labor: Children’s Roles, Care Work, and Familial Obligations in Latino/a Immigrant Families"

_socsci, doi:10.3390/socsci12010036_

Round 1
Reviewer 1 Report
The article addresses a topic that has been little studied in the scientific field, which refers to the participation of children in care work and roles of responsibility in the context of the migration process. This issue appears tangentially in other studies, but here it is explored in depth and this is the great merit of the article.
However, it has some weaknesses that should be corrected.
1. There should be a context from where this study arises, the place (country) and the time in which this review will be made. The article appears to be a "state of the art" or a mini-systematic review on the topic in question, but nothing is indicated in this regard. For example, for the latter, the PRISMA method is used, but (I insist) the methodology is not stated. After reading it, it is known that it was done in the USA and the literature cited is at least 3 decades old. The methodology of this review should be made explicit and justified, now nothing is said about it.
2. The section "Introduction" and "The work of cauidados..." contains repeated ideas, which hinder the reading and subtract space to include new and more interesting ideas.
3. In the introduction, the data of the context of the study should be included, there are no figures on the migration of children and adolescents and if there are any studies that have estimated the time they spend on care work, even if there are figures for the adult world and/or only natives, they could also be compared and it would be interesting.
4. The article is disorganized, it is recommended to use the classic structure: introduction, problem, theoretical framework, methodology, results, conclusions.
5. Figure Nº 1 does not contribute anything, it repeats what is already said in the text, it is suggested to eliminate or complexity, for example, it would be good to include a box with "Examples of domestic tasks, practices, etc. that trace the type of care mentioned".
6. It is suggested to unify the sections that allude to the types of care "Linguistic and cultural support" with "bureaucratic support" and "technological support", because these 3 say similar information.
7. The section "emotional support" should be further elaborated, for example, it should state the sex-gender and age differences within childhood-adolescence.
8. The section "Economic support" should also be deepened because it is revealing as a way of constructing childhood diverging from the typical Western vision of bourgeois childhood.
9. Give more context and details about the DACA program.
Author Response
Thank you for the feedback and the opportunity to strengthen the manuscript. Please see my point-by-point revisions below.
- There should be a context from where this study arises, the place (country) and the time in which this review will be made. The article appears to be a "state of the art" or a mini-systematic review on the topic in question, but nothing is indicated in this regard. For example, for the latter, the PRISMA method is used, but (I insist) the methodology is not stated. After reading it, it is known that it was done in the USA and the literature cited is at least 3 decades old. The methodology of this review should be made explicit and justified, now nothing is said about it.
Thank you for the opportunity to clarify. There is no systematic use of methodology in this paper, given that it is a review piece. Some review pieces do a “state of the literature” and draw on systematic methods to download and capture all existing studies in the topic. These types of reviews are helpful to understand a broad sense of the field, what current research says on a topic, and directions moving forward.
I opted a targeted approach without a systemic methodology because I seek to advance a specific argument and draw on unique studies to support my argument. I do not consider this review piece as a “state of the literature” but rather an essay that expresses an argument and draws on existing empirical examples to support this argument. In my discipline, Sociology, reviews without a methodology are common practice (please see Sociology Compass, as I model the writing in the style of those articles).
- The section "Introduction" and "The work of cauidados..." contains repeated ideas, which hinder the reading and subtract space to include new and more interesting ideas.
I was unable to find the statement “the work of cauidados” in the manuscript but I did re-read the introduction section and tried to condense some of the wording and statements to remove unnecessary items. I also added in (per feedback of Reviewer 2) how the essay relates to the theme of the special issue and why I focus on Latino/a immigrant families.
- In the introduction, the data of the context of the study should be included, there are no figures on the migration of children and adolescents and if there are any studies that have estimated the time they spend on care work, even if there are figures for the adult world and/or only natives, they could also be compared and it would be interesting.
This is a very interesting suggestion. Unfortunately, I was unable to find figures that highlight the frequency of youth’s care work. Any figures available focus on adult women, rather than children. I believe the absence of such figures demonstrates the need to better understand and acknowledge the labor youth can contribute to their families.
- The article is disorganized, it is recommended to use the classic structure: introduction, problem, theoretical framework, methodology, results, conclusions.
I appreciate the suggestion to reorganize the manuscript. However, I am unable to use the suggested format because the review essay draws on specific literature to support the broader argument of youth’s labor. I do not draw on my own empirical data nor do I draw on a specific methodology, but rather I draw on existing research and published material.
- Figure Nº 1 does not contribute anything, it repeats what is already said in the text, it is suggested to eliminate or complexity, for example, it would be good to include a box with "Examples of domestic tasks, practices, etc. that trace the type of care mentioned".
I have removed the figure and replaced it with a new figure that 1) outlines each dimension of support and 2) provides three concrete examples of what this support can look like in practical terms.
Please see the revised table on page 4:
- It is suggested to unify the sections that allude to the types of care "Linguistic and cultural support" with "bureaucratic support" and "technological support", because these 3 say similar information.
I appreciate this suggestion. I elected to keep these three types of support as separate because I believe there is a theoretical benefit in distinguishing and identifying each type of support. For instance, bureaucratic support would not capture the labor youth must do to “google” questions or set up email accounts for parents (i.e., technology and media). I provided further clarification in the manuscript about how youth can (and often do) engage in a combination of these types of care. These types of care can operate differently depending on the youth’s (and parents’) social location.
Please see page 4:
“To uncover the various forms of care work that children partake in to support their immigrant parents, I focus on six dimensions of support: language and cultural, financial, bureaucratic, emotional, legal, and technology. Figure 1 (see below) provides an outline of each type of support and examples to elucidate how the support may take place. I caution the reader to consider these dimensions of support as the major ways youth can help and not as an exhaustive and static list. Youth may employ one or more types of support when supporting family members. For instance, youth may draw on their language skills to construct an email (i.e., language support and technology and media). Youth can also employ their language skills when requesting for social services (i.e., language support and bureaucratic), and those who have undocumented parents may consider how to best ask for services without endangering their family (i.e., language support, bureaucratic, emotional, and legal). The extent of the support depends on the needs of the family and the resources available to the young people.”
- The section "emotional support" should be further elaborated, for example, it should state the sex-gender and age differences within childhood-adolescence.
I have added in a few sentences at the end of this section to clarify that young children may react differently than those who are older: “Younger children may feel ill equipped or uncomfortable to respond to microaggressions while older youth may step in and confront third parties who disregard their loved ones” (see page 6)
When it comes to sex-gender differences, I added in that gender can shape brokering dynamics to the manuscript: “Gender can also affect brokering patterns as daughters and young girls may do more labor in their families (Weisskirch 2010). (see page 5)” To my knowledge, there is very limited work (not much I could find) on gender differences and emotional support. There are certainly gender differences among expectations (i.e., girls are asked to do more domestic care compared to their brothers), but very little work directly examines how emotions differ between girls and boys when brokering. Most work focuses on how the brokering task may produce emotions.
- The section "Economic support" should also be deepened because it is revealing as a way of constructing childhood diverging from the typical Western vision of bourgeois childhood.
I added in a new paragraph in the “financial support” section. The new paragraph now focuses on financial contributions that youth provide to family members who live in their native countries. This paragraph focuses on unaccompanied minors and remittances. This paragraph strengthens the “financial support” section as I demonstrate that youth help their parents in the U.S. and abroad.
The second paragraph is located at the end of page 5:
“Financial support also extends beyond the U.S., as scholars have documented that youth who migrate to the U.S. as unaccompanied minors send remittances to parents and other family members. Unaccompanied youth—many of which are undocumented Latino/a teenagers in low-wage work—feel a sense of urgency to work and send money to loved ones (Canizales 2021, 2022). The urgency to work can position undocumented teenagers to take on exploitative jobs that often hire youth “on the spot.” With a job, unaccompanied youth are able to send remittances and contribute to their family’s survival abroad. For instance, Heidbrink (2018) notes that unaccompanied minors use their remittances to help pay off land debt, purchase food and clothing, and help younger siblings attend better schools. The transitional financial urgency and exploitive work environments negatively impact the health of the undocumented youth workers (Canizales 2022).”
- Give more context and details about the DACA program.
The Deferred Action for Childhood Arrival (DACA) program has been in legal limbo for several years. Recently, the Fifth Circuit of Appeals issued a decision on the legality of the DACA program. I now provide additional context to what has been going on with the program and how it can affect youth (particularly first-time applicants).
The manuscript now contains a footnote to help readers understand that the DACA program allows current DACA recipients to renew their benefits but does not allow for first-time applications.
This footnote is located on page 7:
“The Fifth Circuit Court of Appeals issued a decision on the legality of the DACA program on October 5th 2022. Current DACA recipients are able to continue their benefits and renew their work authorization. However, first-time applications will not be processed at this time (National Immigration Law Center 2022).
Reviewer 2 Report
Thank you for the opportunity to review “Uncovering Youth’s Invisible Labor: Children’s Roles, Care Work, and Familial Obligations in Latino/a Immigrant Families.” This study structure the invisible but vital family roles born by adolescents in immigrant families into categories and provided a literature review of them. The paper is well written and interesting. I offer the following suggestions to help further improve the manuscript:
My main comment relates to the theme of the special issue. I believe the author could do more to explain how the current study relates to mobilities of the children discussed. While a focus on immigrant families relates to the migrant children aspect of the theme, there is currently little understanding of how the study fits with children’s or their families’ mobilities. Perhaps the authors are conceptualizing the immigration event itself as mobility and everything that follows about child care work is related to mobility in this way. If so, could the authors explicitly introduce that connection early in the paper? Or perhaps the author could add more about the implications of care work on a youth’s socio-economic mobility. This could potentially be done by adding a section after section 4 that explores the implications of the dimensions of care work for socio-economic mobility of migrant children. Or another option could be to incorporate these implications within each subsection on the various forms of care work. These are just idea, and I hope the author will in one way or another strengthen the mobilities aspect of the study.
I was initially excited to read line 362, suggesting a discussion of the implications of the study for policymakers. However, the paragraph quickly pivoted away from what policies are needed to better support these youth. Could the author provide more guidance for policymakers about specific lessons learned in this study relevant to policy and how they might begin to address them in a policy setting?
This study is focused on Latino youth only. In the introduction, could the authors provide a rationale for why this population in particular is important to focus on for the study?
Author Response
Thank you for the feedback and the opportunity to strengthen the manuscript. Please see my point-by-point revisions below.
- My main comment relates to the theme of the special issue. I believe the author could do more to explain how the current study relates to mobilities of the children discussed. While a focus on immigrant families relates to the migrant children aspect of the theme, there is currently little understanding of how the study fits with children’s or their families’ mobilities. Perhaps the authors are conceptualizing the immigration event itself as mobility and everything that follows about child care work is related to mobility in this way. If so, could the authors explicitly introduce that connection early in the paper? Or perhaps the author could add more about the implications of care work on a youth’s socio-economic mobility. This could potentially be done by adding a section after section 4 that explores the implications of the dimensions of care work for socio-economic mobility of migrant children. Or another option could be to incorporate these implications within each subsection on the various forms of care work. These are just idea, and I hope the author will in one way or another strengthen the mobilities aspect of the study.
I appreciate the opportunity to clarify. On page 2, I added in a sentence to explain to readers the purpose of the essay in relation to the theme of the overall special issue: “This essay builds on the aims of the special issue by drawing attention to how youth’s labor contributes to the integration and mobility of Latino/a immigrant families in the U.S. and abroad.” (page 2)
The essay also now includes discussion about mobility in section 5 “discussion and conclusion”: “This support is essential to receiving services, learning new skills, and securing opportunities for mobility.” (page 8). Some of the discussions in the “dimensions of support” also highlight unique ways in which youth’s labor contributes to the mobility of the family.
I concretely discuss mobility in the newly added paragraph on “financial support” on page 5. This paragraph (pasted below) helps underscore how financial contributions of undocumented teenagers supports family back in their native countries. The remittances help contribute to the family by paying off dept, purchasing food and clothing, and helping younger siblings attend better schools.
“Financial support also extends beyond the U.S., as scholars have documented that youth who migrate to the U.S. as unaccompanied minors send remittances to parents and other family members. Unaccompanied youth—many of which are undocumented Latino/a teenagers in low-wage work—feel a sense of urgency to work and send money to loved ones (Canizales 2021, 2022). The urgency to work can position undocumented teenagers to take on exploitative jobs that often hire youth “on the spot.” With a job, unaccompanied youth are able to send remittances and contribute to their family’s survival abroad. For instance, Heidbrink (2018) notes that unaccompanied minors use their remittances to help pay off land debt, purchase food and clothing, and help younger siblings attend better schools. The transitional financial urgency and exploitive work environments negatively impact the health of the undocumented youth workers (Canizales 2022).”
In all, this essay aims to capture the labor that youth do in immigrant families and understand how this labor has implications for family mobility as a whole.
- I was initially excited to read line 362, suggesting a discussion of the implications of the study for policymakers. However, the paragraph quickly pivoted away from what policies are needed to better support these youth. Could the author provide more guidance for policymakers about specific lessons learned in this study relevant to policy and how they might begin to address them in a policy setting?
I revised the second to last paragraph in the “discussion and conclusion” on page 9. I do not believe there is one law or a set of laws that fully alleviate youth from engaging in family labor. Rather, I ask readers to consider how seemingly innocuous policies can have an effect on immigrant families—and hence require children of immigrants to set up and help. Consider the following example: during the height of COVID-19 (prior to vaccine availability) hospitals and clinics implemented a “no visitor” rule to stop the spread of COVID-19. While well intended, this policy had negative implications for immigrant families. Children of immigrants were no longer able to join in on visits and parents felt frustrated and dismissed during medical appointments. In this case, it is important to consider how to best support immigrant families while also slowing the spread of COVID-19. My hope with this paragraph is to highlight that not one federal policy will “solve the issue.” However, the needs of immigrant families must be considered in all policies at the local, state, and federal level. If not, youth will still be forced to step up and “bridge” the gaps. I believe my last sentence of the paragraph helps underscore this point in a clearer manner:
“Policies are required to help immigrant families integrate into U.S. society, particularly as the U.S. moves towards a post-COVID-19 society. The COVID-19 pandemic exacerbated the labor children of immigrants do for their parents. For instance, early work suggests that older siblings stepped up to help younger kin with remote learning by providing access to hot spots, translating between parents and teachers, helped parents learn how to create zoom and email accounts, and purchased internet and computers (Author 2020c). Youth also helped their parents navigate healthcare services that transitioned to the web. Immigrant families were forced to work together as they navigated telehealth and circumvented the no visitor rule (then later one visitor rule) in hospitals and clinics (Author 2022c). Other children, particularly those with undocumented parents, may have had to step up more because their parents were ineligible for service programs like the federal Economic Impact Payments (i.e., federal stimulus checks). Policies that support the integration of immigrants will relieve youth from shouldering the responsibility to amalgamate their parents with U.S. society.”
- This study is focused on Latino youth only. In the introduction, could the authors provide a rationale for why this population in particular is important to focus on for the study?
This is an important comment—given that the previous version of the manuscript did not contain a clear statement as to why I focus on Latino/a youth. I now include a statement in the introduction about my decision to focus on this specific population in the last paragraph of the introduction.
Please see page 2:
“This essay focuses on Latino/a families because Latino/as are disproportionally affected by immigration policies and racialized systemic violence (Menjívar & Abrego 2012; Canizales & Vallejo 2021).”